

# Microbial community structure and carbon transformation characteristics of different aggregates in black soil

Danqi Zhao[1,*], Wei Zhang[2,*] and Juntao Cui[1]

[1] College of Resources and Environment, Jilin Agricultural University, Changchun, Jilin, China
[2] College of Modern Agriculture, Chang Chun Polytschnic, Changchun, Jilin, China
* These authors contributed equally to this work.

## ABSTRACT

**Background:** Previous research on whole-soil measurements has failed to explain the spatial distribution of soil carbon transformations, which is essential for a precise understanding of the microorganisms responsible for carbon transformations. The microorganisms involved in the transformation of soil carbon were investigated at the microscopic scale by combining 16S rDNA sequencing technology with particle-level soil classification.

**Methods:** In this experiment,16S rDNA sequencing analysis was used to evaluate the variations in the microbial community structure of different aggregates in no-tillage black soil. The prokaryotic microorganisms involved in carbon transformation were measured before and after the freezing and thawing of various aggregates in no-tillage black soil. Each sample was divided into six categories based on aggregate grain size: >5, 2–5, 1–2, 0.5–1, 0.25–0.5, <0.25 mm, and bulk soil.

**Results:** The relative abundance of Actinobacteria phylum in <0.25 mm aggregates was significantly higher compared to that in other aggregates. The Chao1 index, Shannon index, and phylogenetic diversity (PD) whole tree index of <0.25 mm aggregates were significantly smaller than those of in bulk soil and >5 mm aggregates. Orthogonal partial least-squares discrimination analysis showed that the microbial community composition of black soil aggregates was significantly different between <1 and >1 mm. The redundancy analysis (RDA) showed that the organic carbon conversion rate of 0.25–0.5 mm agglomerates had a significantly greater effect on their bacterial community structure. Moreover, humic acid conversion rates on aggregates <0.5 mm had a greater impact on community structure. The linear discriminant analysis effect size (LEfSe) analysis and RDA analysis were combined. Bradyrhizobium, Actinoplane, Streptomyces, Dactylosporangium, Yonghaparkia, Fleivirga, and Xiangella in <0.25 mm aggregates were positively correlated with soil organic carbon conversion rates. Blastococcus and Pseudarthrobacter were positively correlated with soil organic carbon conversion rates in 0.25–0.5 mm aggregates. In aggregates smaller than 1 mm, the higher the abundance of functional bacteria that contributed to the soil's ability to fix carbon and nitrogen.

**Discussion:** There were large differences in prokaryotic microbial community composition between <1 and >1 mm aggregates. The <1 mm aggregates play an important role in soil carbon transformation and carbon fixation. The 0.25–0.5 mm aggregates had the fastest organic carbon conversion rate and increased significantly more than the other aggregates. Some genus or species of Actinobacteria and

Corresponding author
Juntao Cui, 1597060453@qq.com

Proteobacteria play a positive role in the carbon transformation of <1 mm aggregates. Such analyses may help to identify microbial partners that play an important role in carbon transformation at the micro scale of no-till black soils.

# INTRODUCTION

Soil aggregates are the fundamental building blocks of soil. They can indirectly affect crop yield by regulating the water, air, temperature, and mechanical properties of the soil (*Liu et al., 2021*; *Nyawade et al., 2018*; *Tang et al., 2024*). In addition, soil aggregates is one of the key points in the determination of physical indexes of black soil (*Dong et al., 2023*). Soil mainly consists of various minerals, organic matter, and microorganisms arranged in layers, which are closely linked to the soil's physicochemical properties, soil microorganisms, and soil carbon fractions (*Wang et al., 2023*). As important structural and functional units, soil aggregates extend the spatial heterogeneity of soils, control the activities of microorganisms in the soil space, and influence the supply, conservation, and transformation of soil nutrients (*Li et al., 2023*; *Han et al., 2021*).

Soil organic carbon (SOC) is a major component of the soil carbon pool. Its magnitude can alter the balance between atmospheric $CO_2$ concentrations and soil carbon sink, significantly impacting the global carbon cycle (*Hou et al., 2019*; *Jansson et al., 2021*). SOC is chemically protected in soil microaggregates due to its small particle size. There are more microbial-origin humic substances bound to the adhesive particles of microaggregates, resulting in slow decomposition, which is beneficial for long-term preservation. SOC is also physically protected in macroaggregates; however, there is faster turnover and lower stability. The protection mechanism of organic carbon within each particle-level aggregate is at the core of the carbon fixation process. Nearly 90% of SOC in topsoil is stored in agglomerates, and SOC is a crucial component of soil quality, essential for stable and abundant crop production. Soil organic matter plays a crucial role in plant nutrient cycling and improving soil physicochemical properties (*Wood et al., 2016*; *Zhao et al., 2023*). To comprehend the sequestration mechanisms of soil organic matter, it was necessary to quantify their distribution at the aggregate scale. Humic substances are organic compounds that act as binders, forming stable aggregates. These important substances include humic acid (HA), fulvic acid (FA), and humicin (HM). Since HM is not easily extracted, HA and FA have been studied in greater detail. HA can promote the formation of soil aggregates, and the stability of the soil microaggregates increases with the size of the HA molecule (*Gao et al., 2023*). HA also improves the soil exchange capacity, achieves acid-base balance, improves the soil's water and fertilizer retention abilities, promotes soil microorganism activity, increases aerobic bacteria, actinobacteria, and cellulolytic bacteria populations, accelerates organic material decomposition and transformation, facilitates
nutrient release, and enhances nutrient absorption by crops (*Ampong, Thilakaranthna & Gorim, 2022*; *Li et al., 2019*). Once HA and FA are formed, they may be transformed into each other. In the study of humic substances, HA was higher than FA regardless of the size of the aggregates (*Bongiovanni & Lobartini, 2006*). Among these, microorganisms are the most important components in the formation of humic substances.

Soils are primarily composed of aggregates of different sizes, of which aggregates are composed of hierarchically assembled by minerals, organic matter, and microorganisms. Most soils are primarily inhabited by protozoan consumers, highlighting the significant role of protozoa in energy transfer (*Oliverio et al., 2020*). Several studies have shown that the physicochemical and biological characteristics at the aggregate scale accurately reflect the microbial situation in the soil matrix, which is often overlooked in the study of bulk soils alone (*Wilpiszeski et al., 2019*). During the formation of aggregates, soil microbiota are passively and randomly trapped within the matrix. After formation, the aggregates have different environmental conditions and niches, as well as microbial interactions, including competition and predation, in the surrounding soil or adjacent aggregates (*Fox et al., 2018*). These factors may result in significant differences in the functional aspects, phylogeny, and survival strategies of microorganisms. For example, co-nutrient and fast-growing microorganisms prefer large aggregates rich in unstable carbon, whereas oligotrophic and slow-growing microorganisms tend to reside in nutrient-poor microaggregates (*Trivedi et al., 2017*). Due to sequestration, the evolution and aggregation of microbial communities in different aggregates occur independently (*Wilpiszeski et al., 2019*). As a major player in SOC decomposition, soil microorganisms are a bridge between SOC input and $CO_2$ output. Some studies suggest that changes in soil microbial activity, abundance, and community structure can impact the mineralization of SOC (*Guo et al., 2019*). The structure of the soil microbial community determines and regulates changes in the soil carbon pool. Based on previous research, we know that soil aggregates are fundamental for a comprehensive study of SOC transformation. In the past whole-soil studies, the different proportions of each aggregate in the whole soil and the distinct microbial communities involved in the transformation of substances in each aggregate were not taken into account, despite numerous studies measuring the changes in soil carbon or microbial community structure (*Fu et al., 2019*; *Wang et al., 2023*).

As a consequence, this study aimed to (I) describe the structure of microbial communities in different aggregates of black soils, (II) analyze the correlation between microorganisms and carbon turnover in different aggregates of black soils, and (III) identify microbial partners that play an important role in carbon transformation at the microscale of no-till black soils. To accomplish the aforementioned objectives, we conducted research on the black soil of northeastern China. We proposed three hypotheses: (1) Microbial diversity, composition, and community structures may vary significantly in black soil agglomerates. (2) Black soil aggregates have varying carbon conversion rates, which are associated with the bacteria present on the aggregates. (3) Some bacteria play a crucial role in the transformation of soil carbon in black soil aggregates of varying grain sizes.

## MATERIALS AND METHODS

### Experimental site

Soil samples were collected in Lishu County, Siping City, Jilin Province, China (43°223.32 ″N, 12404′34.33 ″E). The area has a warm-temperate semi-humid continental monsoon climate with high temperatures and rainy summers, cold and dry winters, an annual rainfall of about 600 mm, and an average annual temperature of 6.5 °C. Seasonal freezing occurs from early November to mid to late May of the following year. The soil is classified as black soil (Typic Hapludo-ll, USDA Soil Taxonomy). The black soil contains 35.3% clay, 26.5% silt, and 38.16% sand.

### Experimental design

Three 5 m × 30 m plots were randomly selected in a no-tillage land, and care was taken to avoid marginal areas in order to eliminate edge effects. Sampling took place in late September 2021. According to our previous study, the SOC content decreased significantly at a depth of 20 cm (*Guo et al., 2018*). The sampling was conducted at a soil depth of 0–20 cm in the mentioned plots, following an "S"-type pattern. Surface residues were removed, and a single undisturbed soil block (2,000 cm$^3$) was extracted from a 0–20 cm soil depth using a 25 cm PVC pipe. To determine the moisture content, a 10 g portion of the field-moistened soil was dried in an oven at 105 °C. Soil samples were promptly brought back to the laboratory, where they were cleared of plant debris, small stones, and macrofauna. The soil was then air-dried to 12.3 percent and mechanically sieved. To preserve the integrity of the agglomerates and avoid impacting the microbial composition, the soil was separated by mechanical sieving into >5, 2–5, 1–2, 0.5–1, 0.25–0.5, and <0.25 mm mechanically stable aggregates were then placed in sterile self-sealing bags and marked. In order to investigate the variances in microbial community structure between soil aggregates and bulk soil. Soil samples for our experiment included bulk soil and six soil aggregates. The sieved soil samples were stored in a -80°C ultra-low-temperature refrigerator for 16S rRNA sequencing. Remaining samples were partly used to measure physical and chemical properties before freeze-thaw and partly for freeze-thaw simulation experiments. A total of 48 samples (three replicates × two treatments × seven soil samples).

### Freeze-thaw simulation experiment

Sixteen freeze-thaw cycles are conducted. Seven soil samples were placed in PVC cutting rings for freeze-thaw experiments. The rings were then covered with cling film, and small holes were punched in the film with a needle to maintain aeration. The soil samples used in the freeze-thaw simulation experiment had an initial water content of 12.3%, determined based on information obtained from the field survey and actual temperature. The freezing temperature was set at −15 °C, and the thawing temperature was set at 5 °C, considering the actual local temperature information. After equilibrating the PVC soil columns at 4 °C for 2 h in a thermostatic incubator, the samples were placed at −15 °C for 12 h to simulate the freezing phenomenon. Subsequently, they were kept at 5 °C for 3 days to simulate the thawing phenomenon. The total time for freezing, thawing, and equilibrating was 3 days,

constituting a freezing and thawing cycle. Prior to each freeze-thaw cycle, the samples were weighed and rehydrated with a dropper to reach the initial moisture content specified in the experiment.

## Soil analysis

The soil organic carbon (SOC) content in various aggregates was determined using the potassium dichromate oxidation-spectrophotometric method. Water-soluble organic carbon (WSOC) was determined using the method described by *Yan et al. (2015)*. Humic acid (HA) and fulvic acid (FA) were extracted using the humus composition modification method. The water-soluble organic carbon content of HA and FA was determined using a Shimadzu TOCVCPH instrument (*Duo et al., 2005*). Soil nitrogen availability was assessed using the alkaline hydrolysis diffusion method (*Zhou et al., 2020*).Available phosphate (AP) was extracted using $NaHCO_3$ and then determined using molybdenum antimony spectrophotometry (UV-1800 spectrophotometer; Agilent, Santa Clara, CA, USA). Soil bulk density was determined using the ring knife method (*Hu & Wang, 2020*). The available potassium (AK) was determined using the flame atomic absorption method with a 100 $cm^3$ flame atomic absorption spectrophotometer. The soil pH was determined using a mixture of soil and water with a 0.01 mol·L-1 $CaCl_2$ solution. The water sieving method involves saturating the soil sample for 10 min and then shaking the sieve vertically for 5 min at a frequency of 30 times per minute and an amplitude of 3–50 cm. Finally, the residue on the sieve was collected, dried at 60 °C, and weighed (*An, Darboux & Cheng, 2013*).

## DNA extraction and Miseq sequencing

After following the manufacturer's instructions, the genomic DNA extraction was completed, and the extracted genomic DNA was detected using 1% agarose gel electrophoresis. In PCR amplification, the primers (5′-ACTCCTACGGGAGGCAGCAG-3′) and (5′-GGACTACHVGGGTWTCTAAT-3′) were utilized to amplify the V3-V4 region of the 16S rRNA gene for bacterial community analysis. PCR reactions were performed using the Eppendorf Thermal Cycler Pro S as follows: initial denaturation at 98 °C for 2 s, followed by 40 cycles of denaturation at 98 °C for 10 s, annealing at 60 °C for 30 s, and extension at 72 °C for 1 min, with a final extension at 72 °C for 10 min. The PCR products were amplified and purified using the Agencourt AMPure XP Nucleic Acid Purification Kit after 1% agarose gel electrophoresis. High-throughput sequencing was performed by Ovison Gene Technology Ltd., Melbourne, Australia using the Illumina MiSeq platform.

## Statistical analysis

The Miseq sequencing produced pair-end (PE) double-end sequence data, and the resulting Fastq data underwent quality control processing to generate high-quality FASTA data. The quality of the Fastq data was initially checked using FastQC. Flash and Pear were utilized to eliminate chimeras from FASTA sequences using the VSEARCH method based on the overlap relationship of PE, and the unidentified databases were removed using the

*de novo* method. The short sequences that did not fit the requirements were also removed, resulting in a high-quality valid sequence. The sequences were clustered and annotated with operational taxonomic units (OTUs) using QIIME software at 97% similarity. The vegan package in R (version 4.1.1; *R Core Team, 2021*) was utilized to compute the alpha diversity index, which encompassed the Simpson index, Chao1 index, PD_whole_tree, and the Shannon index. Differences between clusters were analyzed using orthogonal partial least squares discrimination analysis (OPLS-DA). OPLS-DA was performed by R package (ropls 3.1.1). To identify the key microorganisms contributing to differences in community composition among the groups, Linear Discriminant Analysis Effect Size (LEfSe) analysis was conducted using linear discriminant analysis. The vegan package in R (version 4.1.1; *R Core Team, 2021*) was utilized to perform redundancy analysis (RDA) to assess the correlation between key environmental factors and soil bacterial community structure. Additionally, the correlation between dominant phyla and environmental factors was analyzed. A triplicate experiment was conducted, and the results were expressed as the mean ± standard deviation. Data were analyzed using SPSS 22.0 software for one-way analysis of variance. All analytical plots were produced and analyzed by Wekome Bioincloud (https://bioincloud.tech/about-us). The functional abundance of the prokaryotic microbial bacterial communities in each aggregate was predicted using FAPROTAX software (http://www.loucalab.com/archive/FAPROTAX/).

## RESULTS

### Percentage of each aggregate and distribution of organic carbon in black soil before and after freeze-thaw

The impact of short-term freezing and thawing on the soil's capacity in a typical no-till black soil was more complex. The soil bulk weight was 1.31 g/cm$^3$ before freeze-thawing and increased slightly by 0.38% after freeze-thawing, with no significant difference (Table 1). The distribution of various aggregates in the whole soil was determined through soil sieving of adequate soil samples, and the total organic carbon content of both the whole soil and each aggregate was measured. Under mechanical sieving conditions, there was no significant difference in the proportion of 0.5–1 mm aggregates before and after the freeze-thaw experiment. The proportion of aggregates larger than 2 mm was significantly lower ($P < 0.01$), while the proportion of aggregates smaller than 2mm was significantly higher. There was also a significant increase in the proportion of aggregates ranging from 2–5 and 0.25–0.5 mm (Table 2). In contrast, the freeze-thaw experiment did not have a significant effect on the proportion of 1–0.5 mm aggregates, whether in mechanical sieving or wet sieving.

The 2–5 mm mechanically sieved aggregates had the lowest organic carbon compared comparison to the other aggregate sizes before and after the freeze-thaw (Table 3). The ANOVA test revealed that only aggregates of 2–5 and 0.25–0.5 mm showed significant ($P < 0.05$) differences in organic carbon levels before and after freeze-thaw experiments.

**Table 1 Physical and chemical properties of tested soil.**

| Soil | SOC (g/kg-1) | SWC (g/kg-1 ) | AN (mg/kg-1) | AP (mg/kg-1) | AK (mg/kg-1) | Bulk density (g/cm$^3$) |
|---|---|---|---|---|---|---|
| Undisturbed soil | 17 ± 1.55 | 15.32 ± 0.08 | 120.13 ± 0.09 | 60.42 ± 0.18 | 322.43 ± 0.10 | 1.31 ± 0.05 |

Notes:
SOC, soil organic carbon; AK, effective potassium; AN, available nitrogen; AP, available phosphorus; SWC, soil moisture content.
Mean ± standard errors ($n$ = 3).

**Table 2 The proportion of soil aggregates before and after the freeze-thaw simulation experiment.**

| Process | | Proportion of soil aggregates (%) | | | | | |
|---|---|---|---|---|---|---|---|
| | | >5 mm | 5–2 mm | 2–1 mm | 1–0.5 mm | 0.5–0.25 mm | <0.25 mm |
| Mechanical sieving | Pre-freeze-thaw | 40 ± 2.54 | 20 ± 1.46 | 7.8 ± 0.50 | 17.3 ± 0.28 | 5 ± 0.28 | 8.6 ± 0.68 |
| | Post-freeze–thaw | *22.88 ± 1.11* | *14.41 ± 0.61* | **11.72 ± 0.19** | 22 ± 2.06 | **16.33 ± 0.61** | *12.55 ± 0.09* |
| Wet sieving | Pre-freeze-thaw | 0 | 14.31 ± 0.20 | 15.87 ± 0.11 | 8.96 ± 0.09 | 24.09 ± 0.33 | 36.77 ± 0.25 |
| | Post-freeze–thaw | 0 | **20.49 ± 0.23** | *12.1 ± 0.35* | 9.23 ± 0.16 | *26.77 ± 0.19* | **31.41 ± 0.31** |

Note:
Italics indicate significant differences ($p < 0.05$); bold indicates a highly significant difference ($p < 0.01$).

**Table 3 Carbon transformation in freeze-thaw simulation experiment.**

| Soil grain size | PH | SOC (g/kg) | | |
|---|---|---|---|---|
| | | WSA | Pre-freeze-thaw | Post-freeze–thaw |
| Bulk soil | 5.78 ± 0.11a | — | 17 ± 1.55ab | 15.10 ± 0.8cd |
| >5 mm | 5.66 ± 0.02b | 7.92 ± 0.11 | 14.49 ± 3.16b | 15.35 ± 2.86c |
| 5–2 mm | 5.20 ± 0.03d | 13.92 ± 0.17 | 8.83 ± 0.99e | *6.22 ± 0.22e* |
| 2–1 mm | 5.29 ± 0.04c | 12.65 ± 0.27 | 16.20 ± 1.27ab | 12.69 ± 2.16d |
| 1–0.5 mm | 5.05 ± 0.01e | 14.82 ± 0.62 | 11.99 ± 0.91b | 14.21 ± 0.44cd |
| 0.5–0.25 mm | 5.21 ± 0.02cd | 16.78 ± 0.15 | 17.25 ± 1.04ab | *24.23 ± 1.58a* |
| <0.25 mm | 5.17 ± 0.01d | 27.88 ± 0.70 | 19.09 ± 0.91a | 21.54 ± 2.06b |

Notes:
WSA, water-stabilised agglomerate.
Different lower case letters after the numbers in the same column indicate significant differences in the table ($p < 0.05$).
Italics indicate significant differences ($p < 0.05$). Mean ± standard errors ($n$ = 3).

## Microbial community diversity analysis

Soil samples from a total of seven species from whole black soil and six aggregate classes were used. To investigate the species composition of each sample, OTUs were clustered with 97% agreement for all samples and then compared with the database for species annotation of representative sequences of OTUs. Shannon-Wiener curves of the seven soil samples tended to be flat, indicating that the amount of sequencing data was large enough to reflect the most microbial information in the samples. The Chao1 index can analyze the soil abundance, and the higher index indicates the higher bacterial abundance; the Shannon index and PD whole tree index can analyze the diversity of species in soil samples, and the higher index indicates the higher bacterial diversity. The difference in the

**Table 4 Diversity indicators and abundance of bacterial 16S DNA in undisturbed soil and different aggregates.**

| Samples | OTU | chao1 index | Shannon index | PD whole tree index |
|---|---|---|---|---|
| A | 2,618.33 ± 130.91a | 3,702.25 ± 194.72a | 9.42 ± 0.071a | 196.13 ± 11.69a |
| B | 2,570.33 ± 118.94a | 3,730.31 ± 183.36a | 9.42 ± 0.21a | 186.80 ± 7.46ab |
| C | 2,326.67 ± 168.31b | 3,399.90 ± 223.12ab | 9.01 ± 0.30b | 173.81 ± 11.99b |
| D | 2,375.33 ± 124.95b | 3,542.29 ± 182.20ab | 9.11 ± 0.20b | 174.19 ± 7.62b |
| E | 2,322.67 ± 76.51b | 3,433.59 ± 196.62ab | 9.08 ± 0.09b | 176.55 ± 5.15b |
| F | 2,289.33 ± 27.47b | 3,284.66 ± 148.394b | 9.13 ± 0.03ab | 171.50 ± 2.03bc |
| G | 2,216.67 ± 52.56b | 3,292.92 ± 165.88b | 8.96 ± 0.05b | 168.61 ± 4.72c |

Notes:
A, Bulk soil; B, >5 mm; C, 5–2 mm; D, 2–1 mm; E, 1–0.5 mm; F, 0.5–0.25 mm; G, <0.25 mm.
Means followed by the same letter within each column are not significantly different at $P < 0.05$ by Duncan's test. Mean ± standard errors ($n = 3$).

number of OTUs between samples A (bulk soil), B (>5 mm), C (2–5 mm), D (1–2 mm), E (0.5–1 mm), F (0.25–0.5 mm) and G (<0.25 mm) was significant ($P < 0.05$). The Shannon index was ranked from high to low: B>A>D>E>F>C>G; the results for A and B were significantly different from C, D, E and G ($P < 0.05$) (Table 4). The results from the PD whole tree index from high to low were: A>B>E>D>C>F>G; G was significantly different from A; B was ($P < 0.05$) also significantly different from C, D, and E ($P < 0.05$). From the three diversity indices, there were significant differences between A and B aggregate classes and G both in terms of abundance and species diversity.

Analysis under the axes of orthogonal partial least-squares discrimination analysis (OPLS-DA) with two axes of interpretation of PC1 (33.1%) and PC2 (11.5%), respectively, showed that cluster A partially overlapped with clusters B and C, indicating that some species had similar composition. Cluster D was similar in species composition to C; Clusters E, F and G differed significantly from A, B, C, and D in terms of community composition (Fig. 1).

## Differences in relative abundance of bacterial communities

Differences in relative abundance of bacterial communities After filtering out the low quality and short sequence reads from the sequence analysis, a total of 1,629,365 high quality sequences were obtained from 21 samples, with an average of 77,588 reads. After clustering based on the >97% similarity criterion, a total of 5,731 operational taxonomic units (OTUs) were obtained, with an average of 2,388 OTUs per sample. A total of 536 OTU were selected from the total OTU with a relative abundance greater than 0.01%. The OTUs that were selected from the seven samples accounted for 74.81% to 80.31% of the total relative abundance.

Seven major groups of microorganisms were detected by 16S rDNA sequencing, including: Proteobacteria, Actinobacteria, Chloroflexi, Acidobacteria, Gemmatimonadetes, Firmicutes, Bacteroidota, Planctomycetes, *etc.* Among them, Proteobacteria (A: 30.39%, B: 26.24%, C: 32.11%, D: 33.17%, E: 30.21%, F: 29.10%, G: 30.25%), Actinobacteria (A: 19.70%, B: 30.02%, C: 23.73%, D: 25.02%, E: 27.79%,

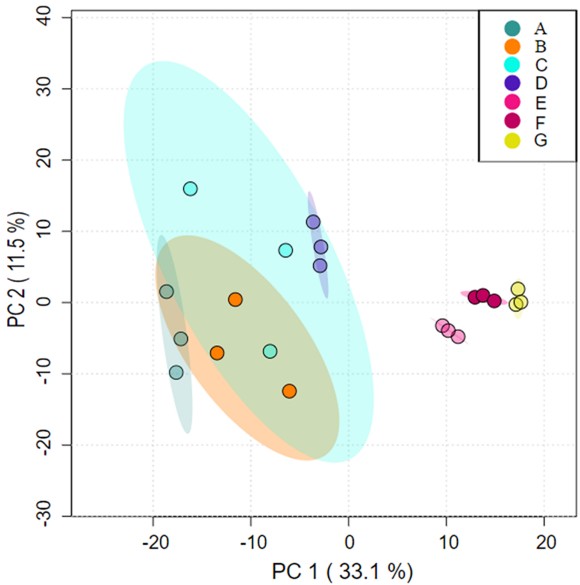

**Figure 1 Differences in microbial community structure between aggregates analysed using orthogonal partial least-squares discrimination analysis (OPLS-DA).** (A) Bulk soil (B) >5 mm (C) 5–2 mm (D) 2–1 mm (E) 1–0.5 mm (F) 0.5–0.25 mm (G) <0.25 mm.

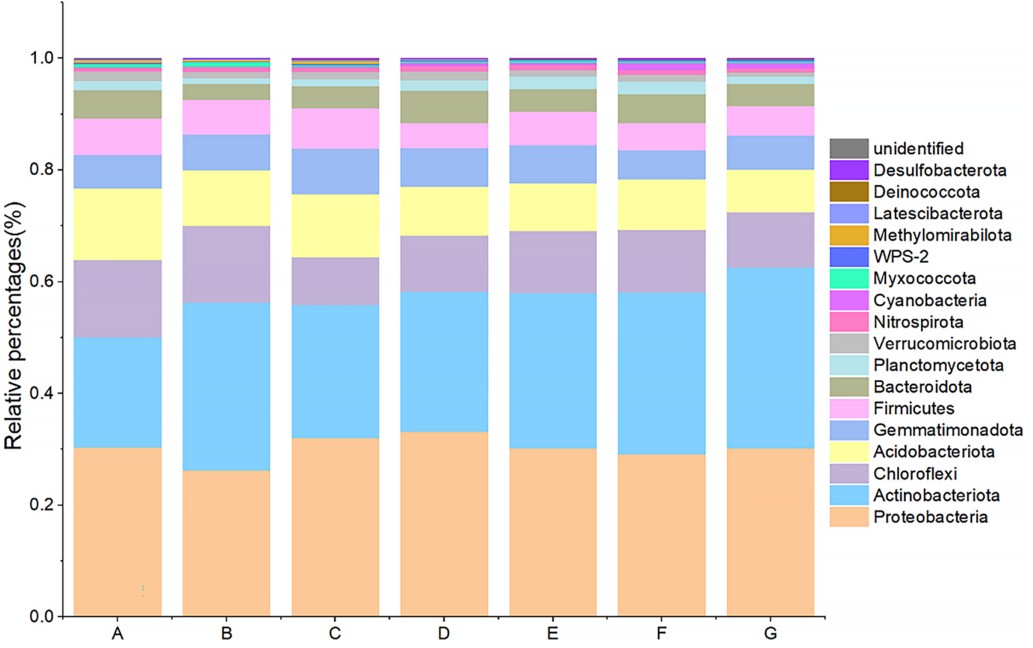

**Figure 2 Bacterial community composition at the phylum level.** (A) Bulk soil (B) >5 mm (C) 5–2 mm (D) 2–1 mm (E) 1–0.5 mm (F) 0.5–0.25 mm (G) <0.25 mm.

F: 28.98%, G: 32.39%), Chloroflexi (A: 13.80%, B: 13.72%, C: 8.62%, D: 10.16%, E: 11.12%, F: 11.25%, G: 9.80%), Acidobacteria (A: 12.84%, B: 9.99%, C: 11.22%, D: 8.73%, E: 8.53%, F: 9.07%, G:7.63%), and Gemmatimonadota (A: 5.99%, B: 6.42%, C: 8.21%, D: 6.94%,
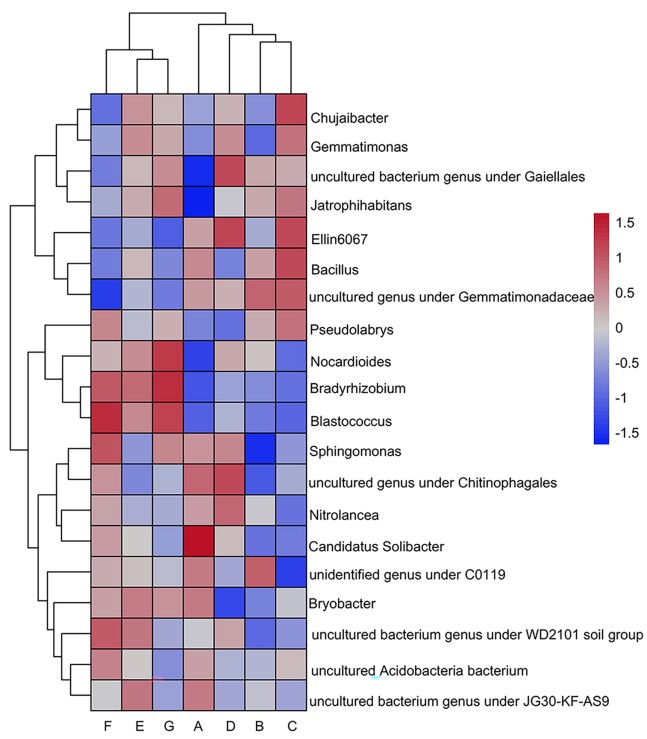

**Figure 3 Clustering of the top twenty genera in relative abundance using heat maps.** (A) Bulk soil (B) >5 mm (C) 5–2 mm (D) 2–1 mm (E) 1–0.5 mm (F) 0.5–0.25 mm (G) <0.25 mm. Darker red represents higher relative abundance and darker blue represents lower relative abundance.

E: 6.82%, F: 5.11%, G:6.18%) accounted for the top four in each group, reaching a total of 75–80% (Fig. 2). The ratio of microorganisms with relative abundance >0.1% of the total microorganisms differed significantly between aggregate samples. Among them, the abundance of Acidobacteria increased significantly with the decrease of aggregate size except for >5 mm; the difference between the lowest abundance of intact soil and other groups ranged from 4.01% to 12.67%. The highest abundance of Acidobacteria was found in the intact soil and the lowest in the <0.25 mm aggregate size, with a difference of 5.22% between the two groups. Chloroflexi had the highest abundance in the intact soils and the lowest in the 2–5 mm class, with a difference of 5.19% between the two groups. Both Planctomycetes and WPS-2 had the lowest abundance of 0.94% and 0.097% in >5 mm aggregate, and the relative abundance was higher in groups E and F. Cyanobacteria had the highest abundance in group F compared to group A by 1.25%.

The heatmap shows the 20 genera with the highest relative abundance at the genus taxonomic level (Fig. 3). The top five dominant genera shared by the seven sample groups were: uncultured bacterial genus under *Gaiellales* (A: 4.43%, B: 6.30%, C: 6.05%, D: 6.87%, E: 5.88%, F: 5.03%, G: 6.16%); *Sphingomonas* (A: 6.03%, B: 3.61%, C: 4.64%, D: 5.88%, E: 4.55%, F: 6.40%, G: 5.77%); Chujaibacter (A: 4.42%, B: 4.11%, C: 7.49%, D: 5.63%, E: 6.12%, F: 3.43%, G: 5.41%) (Fig. 3). The remaining dominating genus in the top five rankings were: *Bacillus* (A: 4.53%, B: 4.22%, C: 4.88%, E: 3.82%, F: 2.87%, G: 2.91%),
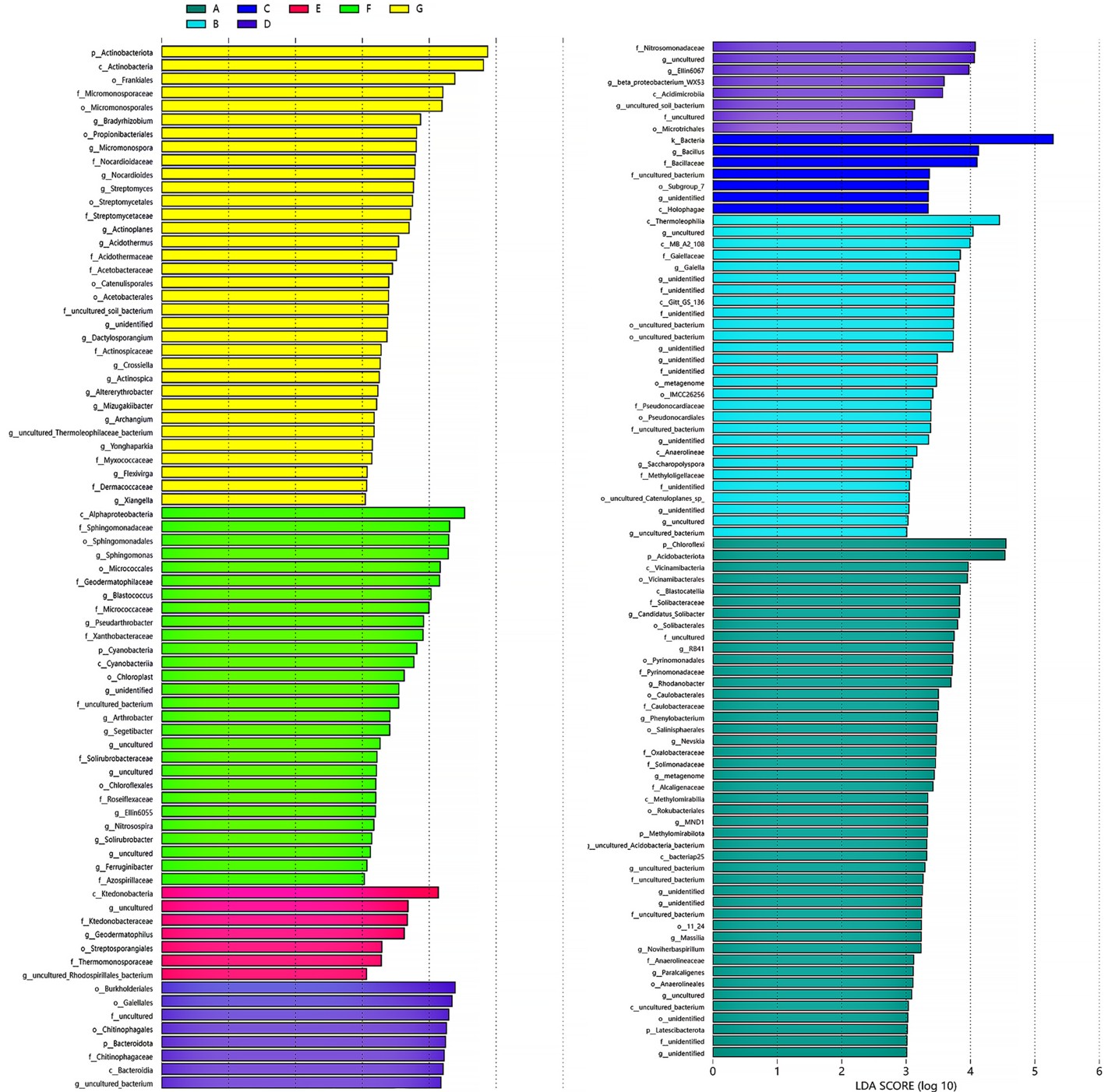

**Figure 4 Biomarkers for LDA effect size analysis of different aggregates to genus level.** (A) Bulk soil (B) >5 mm (C) 5–2 mm (D) 2–1 mm (E) 1–0.5 mm (F) 0.5–0.25 mm. The bars show the species with statistically significant differences in LDA >2d, *i.e.* statistically different biomarkers.

uncultured genera under the order Gemmatimonadaceae (A: 3.70%, B: 4.01%, C: 3.95%, D: 3.38%), uncultured genus under the level of the family Chitinophagaceae (D: 3.50%, F: 3.02%), *Gemmatimonas* (E: 3.12%, G: 2.85%). It is worth noting that Streptomyces (A:

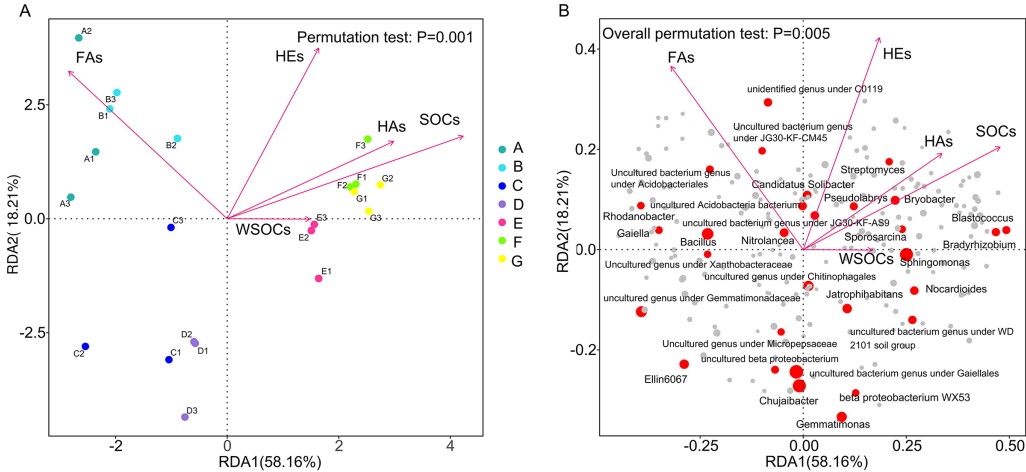

**Figure 5 Redundancy analysis (RDA) demonstrating the response of genus level bacteria to soil environmental factors.** (A) Redundancy analyses (RDA) demonstrating the response of genus level bacterial community distributions to environmental factors; (B) Redundancy analysis (RDA) demonstrating the response of genus level bacteria to soil environmental factors. A, B, C, D, E, F, and G in order represent the structure of the bacterial community in seven aggregates of bulk soil, >5, 5–2, 2–1, 1–0.5, 0.5–0.25, and <0.25 mm. SOC, organic carbon; WSOC, water-soluble organic carbon; FA, fulvic acid; HA, humic acid; HE, extractable humic substances; s, 120-day mean rate of change; Permutation test: *P* = 0.001.            

0.64%,G: 1.63%); *Acidothermus* (A: 0.28%, G: 0.95%); *Blastococcus* (A: 0.35%, G: 2.23%); *Micromonospora* (A: 0.10%, G: 1.56%); *Nocardioides* (A: 0.95%, G: 1.99%); *Geodermatophilus* (A: 0.47%, G: 1.11%); *Jatrophihabitans* (A: 1.20%, G: 2.52%); *Pseudarthrobacter* (A: 0.30%, F: 1.78%, G: 0.98%); *Intrasporangium_flavum* (A: 0.59%, F: 0.96%, G: 0.85%); *Oryzihumus* (A: 0.39%, G: 0.72%); uncultured bacterial genus under the order Gaiellales (A: 4.45%, D: 6.87%, G: 6.16%); all other genus in the Actinobacteria were all less abundant in A than G.

## Analysis of differences in bacterial LESFE across aggregates

LDA effect size analysis determined that there were statistically different biomarkers among multiple groups, *i.e.*, species with significant differences between groups. Species with significant abundance differences in different groups included phylum, order, family, and genus of bacteria. The length of the bar graph represents the effect size of the different species. Some of these genus find their corresponding species through the OTU species annotation tables. *Bradyrhizobium, Micromonospora, Nocardioides, Streptomyces, Actinoplanes, Acidothermaceae, etc.*, were identified as biomarkers for the G group (<0.25 mm) (Fig. 4). *Blastococcus, Pseudarthrobacter, uncultured_Pseudarthrobacter_sp., Arthrobacter (Arthrobacter_sp._DCT-5), Segetibacter, etc.*, were biomarkers in the F group (0.25–0.5 mm). *Geodermatophilus, uncultured Rhodospirillales bacterium, Acidicaldus, Steroidobacter, etc.*, were biomarkers in the E group (0.5–1 mm). At the genus level, there are very few biomarkers at the 2–5 mm cluster and 0.5–1 mm cluster thresholds.

## Correlation between bacteria in different soil aggregates and environmental factors

In the redundancy analysis (RDA) results Red arrows indicate the various physicochemical properties of the soil and longer arrows indicate more significant effects. From Fig. 5A, it can be seen that the RDA1 and RDA2 axes explained 58.16% and 18.21% of the differences in bacterial community structure. RDA analyses showed that there were significant ($P > 0.05$) effects of change rates of soil organic carbon, HE, HA and FA over 120 days on the bacterial community composition at genus level. Soil organic carbon conversion (R2 = 0.74, $P < 0.0005$) and HA conversion (R2 = 0.55, $P < 0.0025$) were positively correlated with the structure of the F and G bacterial communities, and negatively correlated with the structure of the bacterial communities in Groups A, B, C and D. HE conversion rate (R2 = 0.65, $P < 0.0005$) was negatively correlated with C and D. The fulvic acid conversion rate (R2 = 0.70, $P < 0.0005$) was positively correlated with A and B, and negatively correlated with <5 mm aggregates community structure. In the positive direction of the RDA1 axis in Fig. 5B, genus level bacterial abundance of the red solid circles were all positively correlated with soil organic carbon conversion, HA conversion, and HE conversion, respectively, and all negatively correlated with fulvic acid conversion. In the negative direction of the RDA1 axis, genus level bacterial abundance of the red solid circles were all negatively correlated with soil organic carbon conversion, HA conversion, and HE conversion, respectively, and all positively correlated with fulvic acid conversion.

The rate of carbon conversion and the relationship between the absolute abundance of the genus at the taxonomic level of each aggregate were visualized using a correlation heatmap (Fig. 6). *Gaiella* and *Nordlla* were negatively correlated with WSOC conversion ($P > 0.05$). *Lysobacter* abundance was significantly negatively correlated with the fulvic acid conversion rate ($P < 0.01$).Soil organic carbon conversion ($P < 0.001$) and humic acid conversion ($P < 0.01$) were negatively correlated with *Ellin6067* and *MND1* and positively correlated with *Catenulispora* (*Catenulispora*_cavernae) and uncultured genus under the family Rhodobacteriaceae (*Uncultured Frateuria sp.*). The genera that were negatively correlated with carbon transformation were all significant species in the >1 mm aggregates. Among them, *Gaiella*, *Nordlla*, and the uncultured bacterium genus under the order Vicinamibacterales were biomarkers of clusters in >5 mm aggregates. *Uncultured Rhodospirillales bacterium*, and the unidentified genus under the ktedonobacteraceae (*s__uncultured_Frateuria_sp.*) were positively correlated with soil organic carbon transformation rates and were biomarkers on the 0.5–1 mm aggregates. *Blastococcus, P. uncultured_Pseudarthrobacter_sp.* had a positive correlation with the soil organic carbon transformation rate and were biomarkers on the 0.25–0.5 mm aggregates. *Flexivirga* (*s__Flexivirga_sp._M20-45*), *Xiangella* (*s__Micromonospora_phaseoli*), *Actinoplane, Streptomyces, Dactylosporangium*,Catenulispora and *Bradyrhizobium* were positively correlated with soil organic carbon transformation rate and were biomarker on <0.25 mm aggregates. These genera are from Proteobacteria and Actinobacteria.

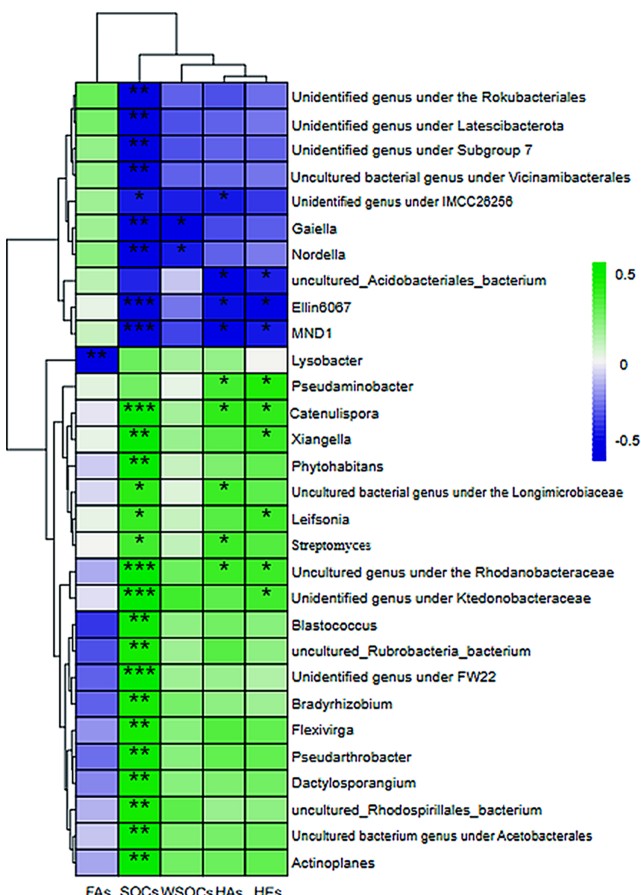

**Figure 6 Heatmap of the correlation between the level of bacterial genus and the rate of change of short-term environmental factors.** (A) Bulk soil (B) >5 mm (C) 5–2 mm (D) 2–1 mm (E) 1–0.5 mm (F) 0.5–0.25 mm (G) <0.25 mm. SOC, organic carbon; WSOC, water-soluble organic carbon; FA, fulvic acid; HA, humic acid; HE, extractable humic substances; s, 120-day mean rate of change; asterisks represents significant correlation, * for <0.05, ** for $P < 0.01$, *** for $P < 0.005$.

**Table 5 The functional abundance of bacterial communities in undisturbed soil and different aggregates.**

| Bacterial community function | Chemoheterotrophy | Aerobic chemoheterotrophy | Aromatic compound degradation | Nitrogen fixation | Cellulolysis function | Manganese oxidation | Nitrate reduction |
|---|---|---|---|---|---|---|---|
| Bulk soil | 2,750.33 ± 61.83aeb | 2,708.67 ± 65.23d | 184.67 ± 28.87c | 132 ± 14.53c | 52 ± 16.09d | 83.33 ± 26.73b | 337.33 ± 29.74ab |
| >5 mm | 2,981 ± 99 de | 2,935.67 ± 71.30d | 312.33 ± 90.56ab | 187.67 ± 56.72bc | 102.33 ± 24.68cd | 117.67 ± 44.84b | 308.67 ± 51.87bc |
| 5–2 mm | 2,849.33 ± 489.35de | 2,781 ± 514.89d | 212.33 ± 60.96bc | 164 ± 25bc | 150.67 ± 81.13bc | 68.67 ± 37.45b | 389.33 ± 6.43a |
| 2–1 mm | 3,230.67 ± 48.84d | 3,142.67 ± 30.37d | 322.33 ± 25.03ab | 211.33 ± 30.92b | 77 ± 26.85bc | 123.67 ± 19.09b | 239.33 ± 46.54cd |
| 1–0.5 mm | 3,929.67 ± 143.26c | 3,795.33 ± 153.44c | 351.33 ± 77.02a | 371 ± 44.14a | 198.67 ± 25.48ab | 229 ± 35.59a | 281.33 ± 36.30bc |
| 0.5–0.25 mm | 4,543.67 ± 124.44b | 4,395.67 ± 101.65b | 326 ± 69.35ab | 384.67 ± 31.66a | 117 ± 21cd | 195.33 ± 15.31a | 207.67 ± 6.66d |
| <0.25 mm | 5,069 ± 273.89a | 4,921.33 ± 259.30a | 402.33 ± 51.40a | 429.67 ± 46.20a | 237.33 ± 55.05a | 214.33 ± 7.51a | 246 ± 48.14cd |

**Note:**
Means followed by the same letter within each column are not significantly different at $P < 0.05$ by Duncan's test. Mean ± standard errors ($n = 3$).

### Differences in bacterial community function among aggregates

Based on the taxonomic analysis of the 16S sequences, the FAPRO-TAX tool was used to annotate the functions of the bacterial community, resulting in the annotation of 28 functional groups. The functions of soil bacteria within different aggregates mainly include chemoheterotrophic chemoheterotrophy (chemoheterotrophy and aerobic chemoheterotrophy), fermentation, chitinolysis, nitrate reduction, aromatic compound degradation, ureolysis, nitrogen fixation, manganese oxidation, iron respiration, *etc.* (Table 5). The dominant functional groups of chemoheterotrophic and aerobic heterotrophic types are most widely distributed in the <0.25 mm particle size. The nitrogen-fixing functional flora and the manganese-oxidizing functional flora were more abundant in tparticles smaller than 1 mm. The abundance of the bacterial community involved in nitrate reduction was higher in the 2–5 mm aggregate size compared to the other aggregates.

## DISCUSSION

In the past, most studies aimed at solving soil problems have focused on the entire soil profile. Analyse the impact of changes in soil carbon content in relation to the structure of the microbial community (*Han et al., 2018*; *Zhang et al., 2021*; *Yang, He & Jia, 2023*). This overlooks the fact that microorganisms involved in carbon transformation are likely to be different at the micro scale. Therefore, we utilized black soil aggregates to investigate the relationship between carbon (organic carbon and humic carbon) conversion rates and bacterial community structure. Further explored the significant effects of certain species or genera on the carbon conversion rates of various black soil aggregates.

There were significant differences in prokaryotic microbial community composition between aggregates smaller than 1 mm and those larger than 1 mm (Fig. 1). This may be due to the fact that microbiological differences in soil aggregates become more pronounced with higher clay content (*Biesgen et al., 2020*; *Wang & Li, 2023*). Microaggregates showed significantly lower levels of both species diversity and species richness compared to *in situ* soils and aggregates larger than 5 mm (Table 4). Interestingly, at the phylum level, the relative abundance of Actinobacteria increased as aggregate size decreased, while microbial richness and diversity decreased with decreasing grain size of the aggregates. The of *Streptomyces*, *Micromonospora*, *Nocardioides*, *Dactylosporangium*, *X. Micromonospora_phaseoli*, and *Actinoplanes* within the Actinobacteria phylum in the <0.25 mm aggregates was significantly greater than in the other particle size aggregates.

In combination with functional annotation of the FAPROTAX microbial community, it was revealed that manganese oxidation was more abundant in <1 mm aggregates. This is because the carbon fixation process depends on the energy provided by the process of manganese oxidation. The connection between the extracellular manganese oxidation process in the metabolic pathway under aerobic conditions for energy storage and the autotrophic $CO_2$ fixation process was further elucidated through transcriptomics (*Yu & Leadbetter, 2020*). The relative abundance of chemolithoautotrophic bacteria was higher in aggregates smaller than 1 mm, suggesting that carbon fixation through Mn oxidation is more intense in these aggregates. Chemoheterotrophy and aerobic chemoheterotrophy

were more prevalent in <0.25 mm than in other aggregates, suggesting that microaggregates are more conducive to promoting carbon metabolism. The highest functional abundance of nitrate reduction activity was observed in 2–5 mm aggregates, which were more conducive to promoting nitrogen metabolism.

In addition, the proportion of particles larger than 2 mm decreased, while the proportion of particles sized between 0.25–0.5 mm increased significantly after the soil was mechanically sieved in freeze-thaw experiments (Table 2). Edwards (2013) also demonstrated that freeze-thaw cycles increased the proportion of 0.5 mm aggregates by 33%. In previous studies, particles ranging from 0.25 to 2 mm were categorized as one grain size. In Li et al. (2023), it was found that the organic carbon storage in the 0.25–2 mm grain size was higher than that in the <0.25 mm grain size (Liu et al., 2023; Li, Qing & Songtao, 2020). However, we further divided the 0.25–2 mm agglomerates into three grain size categories: 1–2, 0.5–1, and 0.25–0.5 mm. There was a significant increase in organic carbon in the 0.25–0.5 mm aggregates before and after freezing and thawing, compared to the other aggregates (Table 3). The freeze-thaw treatment of mechanically sieved black soils resulted in a significant increase in the proportion of macroaggregates and organic carbon content (Zhou et al., 2020). The formation of soil aggregates could inhibit the decomposition of organic matter and promote the fixation of soil carbon. Therefore, it is possible that 0.25–0.5 mm agglomerates are important sites for carbon fixation in black soil.

The redundancy analysis revealed a positive correlation between organic carbon conversion on <1 mm aggregates and prokaryotic microbial communities (Fig. 5). Most soil microorganisms live in cyclically interconnected communities closely associated with soil aggregates. These communities are strongly constrained by clusters of minerals and organic carbon in aggregates smaller than 2 mm, which persist through mechanical disruption and wetting events (Wilpiszeski et al., 2019). The aggregates smaller than 1 mm may be significant locations for the transformation of soil organic carbon. Soil organic carbon plays a crucial role in shaping microbial community composition (Teng et al., 2021; Ma et al., 2023), potentially resulting in variations in microbial communities involved in organic carbon transformation across different soil aggregates. Among the differential biomarkers between groups, combined with the rate of conversion of SOC and the correlation of microbial genera at the genus level, it was found that the unidentified genus under Ktedonobacteraceae and the uncultured Rhodospirillales bacterium were significantly and positively correlated with the SOC transformation rate in 0.5–1 mm aggregates. Blastococcus, P. uncultured_ Pseudarthrobacter_sp.,the uncultured bacterium genus under Acetobacterales were found to have a significant and positive correlation with the transformation rates of SOC in 0.25–0.5 mm aggregates. Bradyrhizobium, uncultured Rubrobacteria bacterium, Actinoplane, Streptomyces, Dactylosporangium, F. Flexivirga_sp. _M20-45, X. Micromonospora_phaseoli, unidentified genus under the ktedonobacteraceae (s__uncultured_Frateuria_sp.), C. Catenulispora_cavernae were significantly and positively correlated with the transformation rates of SOC in aggregates smaller than 0.25 mm.

The genera identified above, which positively influence the rate of organic carbon conversion, mostly belong to the phylum Actinobacteria. Actinobacteria, as r-strategists, are microorganisms associated with negative excitation effects. They tend to thrive in soils with lower effective carbon content and prioritize the use of plant residues, which contributes to the accumulation of SOC (*Fu, Song & Li, 2022*). Gram-positive bacteria include more oligotrophic microorganisms that preferentially utilize more recalcitrant carbon as a microbial carbon source, and oligotrophic microorganisms grow slowly. Gram-negative bacteria utilize more readily decomposable carbon sources. As a result, the microbial community structure of different aggregates varied in the efficiency of carbon source utilization. This is one of the reasons why the RDA analysis showed a positive correlation between the genus under the phylum Actinobacteria and SOC conversion in the <1 mm soil aggregates. Actinobacteria are widely recognized as an extremely valuable microbial resource, contributing 45% of biologically active substances in microbial resources. They have promising prospects for development and application (*Singh, Young & Silver, 2017*). *Dactylosporangium*, *X.Micromonospora_phaseoli* belongs to the family Micromonosporaceae, which produces alkaloids, aminoglycosides, and other antibiotics (*Hifnawy et al., 2020*) that inhibit Gram-positive bacteria or viruses (*Al-Ansari et al., 2019*). Streptomycetaceae is the most extensively researched family within Actinobacteria, with *Streptomyces* contributing approximately 80% of the total secondary metabolites of actinomycetes and serving as the primary source of various antibiotics (*Alam et al., 2022*). *Streptomyces* isolated from various sources have been found to be beneficial for organic decomposition and promoting plant growth (*Passari et al., 2016*). In addition, *Streptomyces* can induce defense pathways in the host plant, enabling it to respond more rapidly to pathogen attacks (*Gao et al., 2021*). Actinobacteria, Proteobacteria, and Bacteroidetes are also considered potential decomposers (*Buresova et al., 2019*). *Streptomyces* is essential for the carbon cycle, and members of this genus are involved in the decomposition of insoluble organic debris (*Ventura et al., 2007*). *Bacillus* can break down and utilize cellulose and proteins in the soil (*Kumar, Khare & Dubey, 2012*; *Nevins, Nakatsu & Armstrong, 2018*).

Brazilian soils were analyzed using spectroscopy under no-tillage conditions, and humic acids were characterized based on the degree of aromaticity (the ratio of humic substances E465 to E665 extracted by UV/visible spectrophotometry). It was found that <2 mm aggregates were much more dependent on the degree of aromaticity than the >2 mm aggregates (*Machado et al., 2020*). These results were consistent with the findings of our study. The degree of aromaticity depends on the extent of humification, as microbiological activity releases nitrogenated compounds that react with lignin (*Stevenson, 1994*). Microorganisms play a crucial role in the formation of humic and fulvic acid structures (*Silva et al., 2014*). Organic matter, such as lignin and polysaccharides, is degraded by microorganisms to form complexes. For instance, lignin-carbohydrate complexes have a tendency to form aggregates, particularly mycelial aggregates (*Zeng & Cheng, 2002*). Several studies have shown that the application of humic acid can result in a significant increase in soil Actinobacteria (*Xue, Xi & Wang, 2016*; *Song et al., 2019*). Research has also demonstrated that actinobacteria can fully utilize cellulosic material, and

their impact on the structure of humic acid is significantly greater than that of other types of microorganisms (*Li et al., 2016*). The conversion rates of humic acid in RDA analysis were positively correlated with the bacterial community structure in aggregates smaller than 1 mm (Fig. 5). Differential biomarkers of difference between the groups, combined with humic acid conversion rate and genus-level microbial correlation, showed that the unidentified genus under the ktedonobacteraceae (*s__uncultured_Frateuria_sp.*), *P. uncultured_ Pseudarthrobacter_sp.* were significantly and positively correlated with the conversion rate of humic acid in 0.25–0.5 mm macroaggregates. *Streptomyces and X. Micromonospora_phaseoli* were significantly positively correlated with the conversion rate of humic acid in aggregates smaller than 0.25 mm. This may also explain the prediction of bacterial community function. The relative abundance of cellulolysis was significantly higher in aggregates smaller than 0.5 mm than in those larger than 2 mm, while the relative abundance of aromatic compound degradation was significantly higher in aggregates sized 0.5–1 mm and smaller than 0.25 mm than in aggregates larger than 2 mm.

## CONCLUSIONS

This study assessed black soils at the agglomerate level to identify variations in microbial community structure and to explore the potential implications of these differences for carbon transformation. There were significant differences in prokaryotic microbial community composition between aggregates smaller than 1 mm and those larger than 1 mm. The aggregates measuring less than 1 mm play an important role in the transformation of soil carbon and its fixation. The conversion rate of HA was faster in aggregates smaller than 0.5 mm. Fulvic acid exhibited a higher conversion rate on aggregates larger than 5 mm. The 0.25–0.5 mm aggregate showed the fastest organic carbon conversion rate and a significant increase compared to the other aggregates. Certain genera or species of Actinobacteria and Proteobacteria play a beneficial role in the carbon transformation of aggregates smaller than 1 mm. Future research should investigate the potential use of bacteria that have demonstrated a positive impact on carbon sequestration in this study at the micro scale. This could enhance the quality of black soils and contribute to the restoration of degraded black soils.

## ACKNOWLEDGEMENTS

We thank Mr Dowson for his guidance on this thesis and Zhengyi Gu for his help with the experiments.

### Funding

This work was supported by the National Key Research and Development Program of China (SQ2023YFD15000, No. 2017YFD0200601) and the Science and Technology Department of Jilin Province (No. 20210204170YY). The funders had no role in study design, data collection and analysis, decision to publish, or preparation of the manuscript.

## Grant Disclosures

The following grant information was disclosed by the authors:
National Key Research and Development Program: SQ2023YFD15000, No.
2017YFD0200601.
Science and Technology Department of Jilin Province: 20210204170YY.

## Competing Interests

The authors declare that they have no competing interests.

## Author Contributions

- Danqi Zhao conceived and designed the experiments, performed the experiments, analyzed the data, prepared figures and/or tables, and approved the final draft.
- Wei Zhang analyzed the data, authored or reviewed drafts of the article, materials, and approved the final draft.
- Juntao Cui conceived and designed the experiments, authored or reviewed drafts of the article, and approved the final draft.

## Data Availability

The data is available at NCBI SRA: PRJNA1035756.

## Supplemental Information

Supplemental information for this article can be found online at http://dx.doi.org/10.7717/peerj.17269#supplemental-information.

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
