# Peer review of "Microbial community structure and carbon transformation characteristics of different aggregates in black soil"

_PeerJ, doi:10.7717/peerj.17269_

## Round 0.1 · original submission · Major Revisions

While doing your revisions, please pay more attention to Statistical Analyses and redundancy as pointed out by the reviewers.

Reviewer 1 ·

Basic reporting

The manuscript entitled 'Microbial Community Structure and Carbon Transformation Characteristics of Different Aggregates in Black Soil' is interesting and fits within the aims and scope of PeerJ. The authors are advised to make some major changes throughout their manuscript before final acceptance. Please see below for specific comments:

1. Minor corrections of the English language (e.g., typos, syntax errors, etc.) are necessary throughout the manuscript. Overall, the manuscript is well-written.
2. Abstract: Please mention the specific name of the sequencing technology used instead of the 'high throughput sequencing technology' (L 19-20). Also, please delete the underscore "_" (L 36).
3. Line 63: Replace 'CO2' with 'CO2'.
4. Lines 87-105: Use 'aggregates' where necessary.
5. Please rephrase Lines 106-117. I expect a citation of your previous study in Lines 107-110.
6. Statistical analyses: Although Trimmomatic and Pear are proficient at quality trimming and merging paired-end reads, authors are suggested to use FastQC for the initial check. For chimera detection and removal, while the uchime method is effective, authors could utilize DADA2 or VSEARCH, considering their effectiveness in detecting chimeras in complex microbial communities. Importantly, these tools offer refined algorithms for a more precise analysis. For statistical analysis, Excel and Origin may not be appropriate considering the complex nature of your data. R, with its extensive libraries specifically designed for ecological and microbiome data analysis (such as vegan and phyloseq), would be an appropriate choice. This could enhance both the depth and breadth of your statistical analysis and interpretation.

7. Figure 2: Please replace 'relative percentages' with 'relative abundance (%)'. Would it be possible for you to recreate the figure to report the top 20 phyla? That would be interesting.
8. Figure 3: Please report the top 20 genera.
9. Figure 4: Please revise it.
10. Authors are suggested to improve the discussion of their findings.

Good luck!

Experimental design

no comment

Validity of the findings

no comment

Reviewer 2 ·

Basic reporting

Language is not in most of the sections and needs to be reworked.

Experimental design

It is really hard to understand the experimental design. Too much unnecessary information here. Please rework on it and provide a concise experimental design. The experimental design, treatments used, number of replications, and the reasons for choosing this approach should be sufficient.

Validity of the findings

The flow is not smooth. Authors should clearly set their objective and other flows should direct to this objective.

Additional comments

Abstract:
The results part needs revision.
• Line 28….” from” is missing following …” different”.
• Line 30, You mentioned partial least square…which type? There are several partial least analysis methods, please be specific and mention the specific method you used in this study.
• line 34-42. Many confusing sentences. Please put the findings in a more concise way, in short sentences.
• Please add a concluding remark and/or recommendation.
Introduction
More than 50 percent of the literature cited was at least 10 years old. Please provide citations from recent publications where possible.
• Line 58-61 please provide citation for this.
• Line 63, in CO2, 2 should be in subscript format, here and elsewhere.
• Line 107, which study? If this is part of a study you previously conducted, please cite the previous work.
• Line 115-117, remove this sentence and add your hypothesis and objective.
• Line 176, please provide the original citation Jones et al.
Results and conclusions
• Figure 5. Agglomerates were constraining factors in your study since they were categorical variables. Consequently, no correlation was possible to be conducted with categorical and quantitative variables with RDA. So, if there is an interest in the effect of the agglomerates reflected in the analysis results, you may need to run partial RDA, so that the variance captured with the analysis may also be expected to be greater than what the current one. If interested to keep the results, the title needs to be corrected accordingly.
• Redundancy analysis (RDA) demonstrating the response of bacteria to soil environmental factors.
• Conclusions should also provide the take home message and future research directions.

---

## Round 0.2 · Minor Revisions

Please revise and resubmit. A few minor revisions are requested.

Reviewer 1 ·

Basic reporting

The authors have revised their manuscript properly and responded to all raised concerns. I suggest acceptance of this manuscript.

Experimental design

No comment

Validity of the findings

No comment

Additional comments

No comment

Reviewer 2 ·

Basic reporting

Authors may need to make minor revision to the manuscript before it will be accepted. Some sentences are incomplete. Abbreviations need to be defined the first time they are mentioned in the manuscript.

1. Line 18-20: The sentence “The microorganisms involved in the 19 transformations……” should be taken to under the methods in the abstract.
2. Please define the shorts like the RDA, PD, etc., the first time they appear.
3. Line 27, was the relative abundance greater or lower? Mentioning significant difference alone is not enough here.
4. Line 52-53, needs citation.
5. Line 222 looks like something is missing.
6. Line 341, change ‘analyzed’ to ‘visualized’.

Experimental design

Previous issues are addressed in the current version of the manuscript.

Validity of the findings

The findings are valid as the study was well designed and findings were based on the results obtained from valid sample size and powerful statistical tools used.

---

## Round 0.3 · accepted · Accept

Thanks for the revisions. A positive recommendation has been made.